# High Agreement between Barrett Universal II Calculations with and without Utilization of Optional Biometry Parameters

**DOI:** 10.3390/jcm10030542

**Published:** 2021-02-02

**Authors:** Yakov Vega, Assaf Gershoni, Asaf Achiron, Raimo Tuuminen, Yehonatan Weinberger, Eitan Livny, Yoav Nahum, Irit Bahar, Uri Elbaz

**Affiliations:** 1Department of Ophthalmology, Rabin Medical Center, Petah-Tikva 49100, Israel; yakovvega@mail.tau.ac.il (Y.V.); Assafge@clalit.org.il (A.G.); asaf.achiron@uhbristol.nhs.uk (A.A.); Yehonatanwe@clalit.org.il (Y.W.); EitanLi@clalit.org.il (E.L.); yoavn@clalit.org.il (Y.N.); iritba2@clalit.org.il (I.B.); 2Sackler Medical School, Tel Aviv University, Tel Aviv 6997801, Israel; 3Helsinki Retina Research Group, University of Helsinki, 00100 Helsinki, Finland; raimo.tuuminen@helsinki.fi; 4Department of Ophthalmology, Kymenlaakso Central Hospital, 48210 Kotka, Finland

**Keywords:** Barrett Universal II formula, intraocular lens calculation, axial length, cataract surgery

## Abstract

Purpose: To examine the contribution of anterior chamber depth (ACD), lens thickness (LT), and white-to-white (WTW) measurements to intraocular lens (IOL) power calculations using the Barrett Universal II (BUII) formula. Methods: Measurements taken with the IOLMaster 700 (Carl Zeiss, Meditec AG, Jena, Germany) swept-source biometry of 501 right eyes of 501 consecutive patients undergoing cataract extraction surgery between January 2019 and March 2020 were reviewed. IOL power was calculated using the BUII formula, first through the inclusion of all measured variables and then by using partial biometry data. For each calculation method, the IOL power targeting emmetropia was recorded and compared for the whole cohort and stratified by axial length (AL) of the measured eye. Results: The mean IOL power calculated for the entire cohort using all available parameters was 19.50 ± 5.11 diopters (D). When comparing it to the results obtained by partial biometry data, the mean absolute difference ranged from 0.05 to 0.14 D; *p* < 0.001. The optional variables (ACD, LT, WTW) had the least effect in long eyes (AL ≥ 26 mm; mean absolute difference ranging from 0.02 to 0.07 D; *p* < 0.001), while the greatest effect in short eyes (AL ≤ 22 mm; mean absolute difference from 0.10 to 0.21 D; *p* < 0.001). The percentage of eyes with a mean absolute IOL dioptric power difference more than 0.25 D was the highest (32.0%) among the short AL group when using AL and keratometry values only. Conclusions: Using partial biometry data, the BUII formula in small eyes (AL ≤ 22 mm) resulted in a clinically significant difference in the calculated IOL power compared to the full biometry data. In contrast, the contribution of the optional parameters to the calculated IOL power was of little clinical importance in eyes with AL longer than 22 mm.

## 1. Introduction

In cataract surgery, the demand for excellent post-operative visual and refractive outcomes is increasing, along with patients’ expectations for spectacle independence [1,2]. As such, predicting the optimal intraocular lens (IOL) power that will give the best postoperative result and avoiding postoperative refractive surprises is of utmost importance, both for patient satisfaction and for surgeon reputation [3].

Amongst the new generation of IOL formulas, the one regarded as the most accurate and predictable is the Barrett Universal II (BUII) formula [4,5,6,7,8,9]. The BUII formula utilizes optional parameters recommended to refine accuracy, including anterior chamber depth (ACD), lens thickness (LT), and white-to-white (WTW) measurements [10]. It may be, however, challenging to acquire all of the optional parameters in some of the patients requiring surgery and the measurements may not be possible in all centers due to the limitation of suitable devices [11]. Furthermore, IOL power calculation with the BUII utilizing partial biometry data (axial length (AL) and keratometry (K) only) is still widely in use [12].

Currently, it remains unclear whether incorporating optional parameters yields a clinically significant difference in the calculation of the predicted IOL power. Here, we aimed to assess the contribution of each optional parameter to the IOL calculation.

## 2. Methods

### 2.1. Study Design

Biometry measurements of consecutive patients, who underwent cataract surgery in a tertiary medical center between January 2019 and March 2020, were obtained with the IOLMaster 700 (Carl Zeiss Meditec, Jena, Germany). The collected data included the following biometry information: AL, K, ACD, LT and WTW. IOL power with an intended emmetropic refractive outcome was then calculated by using all the variables of the BUII formula. Thereafter, multiple IOL power calculations were performed, each time omitting optional parameters (ACD, LT, WTW) one by one in a different possible combination using the online BUII formula calculator provided by the Asian-Pacific association of cataract and refractive surgery; APACRS (http://calc.apacrs.org/barrett_universal2105/). This study adhered to the tenets of the Declaration of Helsinki and was approved by the Ethics Committee of the Rabin Medical Center.

### 2.2. Visual-Basic Software for Applications Code

To increase the accuracy of our calculations and prevent human error, a Visual-Basic Software for Applications code that automatically inserts the data into the online calculator was used. The code was checked and compared to manual typing. Data of all patients were inserted eight times into the calculator; one time with all parameters and seven times with different combinations of optional parameters.

### 2.3. Axial Length Sub-Analysis

The primary outcome measure was to compare each of the predicted IOL power aiming for emmetropia granted through each calculation method, to the one that employed all parameters in its estimate. Sub-analysis comparing the resulting IOL power stratified by AL was carried out. Group I consisted of eyes with AL ≤ 22 mm, Group II was composed of eyes with AL greater than 22 mm but smaller than 26 mm, and Group III included eyes with an AL ≥ 26 mm [13].

### 2.4. Power Analysis

Sample size calculation was performed using the mean IOL power (19.50 ± 5.11 D) emerging from the 501 measurements, using a freely available power calculator (ClinCalc.com). We estimated a 3% difference between the mean IOL power generated by all biometry data and the expected mean IOL power generated by the partial biometry data. A sample size of 324 eyes would be required to detect such a difference between the full BUII data and the partial formula, with a significance level of 0.05 and a power of 80%.

### 2.5. Statistical Analysis

Data were analyzed using SPSS software (SPSS for Windows, Version 26.0. IBM Corp, Armonk, NY, USA). Distribution normality of continuous variables was performed using the D’Agostino-Pearson test. A paired *t*-test was used to compare the IOL diopteric power by BUII formula including all parameters to those calculations with variation of parameters. Correlation analyses were performed between the mean difference of the partial and complete BUII calculations and the AL and K readings (Spearman’s rho rank correlation). A value of *p* < 0.05 was considered statistically significant.

## 3. Results

### 3.1. Clinical Characteristics

The study group consisted of 501 right eyes of 501 patients. The mean age was 65.2 ± 17.4 years and the mean AL was 23.75 ± 1.88 mm (Table 1). Group I (AL ≤ 22 mm) consisted of 94 eyes (18.8%), Group II (AL greater than 22 mm and smaller than 26 mm) was composed of 324 eyes (64.7%), and Group III (AL ≥ 26 mm) included 83 eyes (16.6%).

### 3.2. Agreement between Partial and Full Biometry Data in Eyes with Short, Intermediate and Long Axial Length

The mean IOL power for emmetropia while using the BUII formula with all parameters available was 19.50 ± 5.11 diopters (D). When comparing it to the other seven combinations, reflecting partial biometry data, the maximal mean absolute difference was 0.14 ± 0.12 D (*p* < 0.001, Figure 1 and Table 2). Stratification of data according to AL showed that the optional variables had the least effect in longer eyes with AL ≥ 26 mm (mean absolute difference in IOL power ranged from 0.02 to 0.07 D, Figure 2), while they had the greatest effect in short eyes with AL ≤ 22 mm (mean absolute difference in IOL power ranged from 0.10 to 0.21 D, Figure 2).

The percentage of eyes with a mean absolute difference of ≥ 0.25 D in the IOL power calculated between selected partial and all biometry data, stratified by axial length, is represented in Figure 3. The short AL group showed the highest proportion of eyes (32.0%) with this difference when using AL and keratometry data only. The intermediate and long AL groups demonstrated a proportion of 13.0% and 2.0% of eyes, respectively (*p* = 0.017 for the intermediate AL and *p* = 0.024 for the long AL, when comparing to the short AL).

Similarly, a smoothed line graph [4] of non-absolute difference in the calculated IOL power between selected partial and all data biometry analyzed in accordance with AL, confirms the higher difference in short eyes in Figure 4.

## 4. Discussion

The accuracy of postoperative predicted refraction following cataract extraction surgery is constantly increasing with the use of a new generation of IOL formulas. As opposed to earlier generations of formulas that mainly utilize AL and K readings, for the prediction of postoperative refraction, newer formulae such as the BUII use other factors that take into account the effective lens position of the implanted IOL, which vary between different AL [8]. While the technology in the field of ocular biometry prior to cataract surgery has significantly advanced in recent years [14], the accuracy of the postoperative refractive outcome is limited by the biometry data that a specific IOL calculation formula utilizes [15]. The purpose of this study was to examine the contribution of the three additional biometry parameters (ACD, LT, and WTW) to the final IOL power calculation, when employing the BUII formula, and hence, assess which of these optional parameters is mandatory for achieving the best reliable result. Here we show that ACD, LT, and WTW, optionally used by the Barrett calculator for enhancing accuracy, have more clinical significance in eyes shorter than 22 mm.

The BUII formula that incorporates in its calculation AL, K readings, ACD, LT, and WTW measurements is considered an accurate and reliable formula for all AL [5,6,7]. The creators of the formula recommend using all parameters for the best results, though in some cases, it might be difficult to acquire all of them. Such is the case with bedridden patients, the mentally disabled, and in the pediatric population undergoing cataract extraction surgery, where standard portable devices are utilized for ocular biometry, with mainly AL and K readings provided by these machines. In addition, when considering a secondary IOL implantation in an aphakic eye, ACD and LT cannot be measured even with sophisticated cutting-edge technology table-mounted machines. Furthermore, advanced biometry machines are not always in reach. For these reasons, our study estimated the importance of each of these optional parameters when utilizing the BUII formula. By omitting alternately one or two or all three optional parameters we could show that adding the LT, ACD and WTW variables to the BUII calculation is of more clinical importance in eyes shorter than 22 mm, and less so in longer eyes. Throughout the entire cohort, the maximal mean absolute difference from the IOL power calculated with all parameters was 0.14 D. While this difference was statistically significant, it is clinically small and considered as a good acceptable refractive outcome [3,16]. To the best of our knowledge, there is only one study showing the accuracy of IOL power calculation formula in the presence of partial biometry. In their study from 2013, Srivannaboon et al. found that LT was of little value when employing the Holladay II formula [17]. They showed that the median absolute difference of postoperative spherical equivalent refraction and predicted postoperative spherical equivalent refraction, with or without the LT parameter, was only 0.02 D, regardless of AL. Similarly, in our study, when examining all possible different calculation methods, we also found that the LT parameter is of no significance, unless it is combined with ACD. We assume that this might be due to the fact that the BUII uses a fix ratio between ACD and LT to project the effective lens position, thus making the solo variable of LT insignificant. Similarly, incorporating WTW measurements had little to no effect on the resulted IOL power when used solely or in conjunction with LT. The most important parameter among the optional parameters was found to be ACD, as when it was not employed in the calculation of the BUII, results were identical to calculations based only on the AL and K readings. Instead, when ACD was utilized, the IOL power calculation was similar to that generated by all parameters with only 0.05 to 0.11 D mean absolute difference between them.

When observing the effect of the optional variables in relation to AL, it seems that they have little clinical significance when the AL is greater than 22 mm. The maximal mean absolute difference between all biometry data and when only AL and K readings were used in Groups II (AL greater than 22 mm and smaller than 26 mm) and III (AL ≥ 26 mm) was only 0.14 and 0.07 D, respectively. It is important to note that while these differences were statistically significant, they too have little clinical importance. When examining the eyes in Group I (AL ≤ 22 mm), the significance of using all optional parameters seems greater, as a maximal mean absolute difference of 0.21 D was found, when compared to utilizing AL and K readings only. Similarly, this group demonstrated the highest proportion of eyes with an absolute difference of ≥0.25 D between all biometry data and when only AL and K readings were utilized. This seems reasonable, as the predicted refractive error produced by an error in the effective lens position estimate has a greater effect in eyes with short AL [18]. This group of eyes can therefore benefit from the optional parameters proposed by the BUII to achieve a more precise outcome. It is also of note that this small group swayed the results of the entire cohort, and the overall clinically negligible small differences found were even smaller if only eyes with AL of 22 mm or longer were examined. Indeed, when repeating the analysis with Groups II and III only (*n* = 408), the mean difference between all biometry data and that of with AL and K readings only was 0.01 D and the mean absolute difference was 0.12 D.

It is worth mentioning the limitations to our study, with regard to the relatively small sample size of group I (*n* = 94) and group III (*n* = 83), which comprised eyes with an AL ≤ 22 mm and ≥26 mm, respectively. As such, and even though the results were statistically significant, the repeatability of the results might be even better if the sample size was larger. Nevertheless, a limitation by the small sample sizes of eyes with short AL was noted in previous studies as well [5]. Furthermore, postoperative refractive data and prediction errors were not available and therefore we could not assess how the change in IOL power selected would influence the accuracy of the BUII. Of note, investigating the accuracy of the BUII was beyond the scope of our study and was shown previously in numerous studies.

In conclusion, in this study, we attempted to examine the effect of the optional parameters (ACD, LT, WTW) on the IOL power selected prior to cataract surgery, through the BUII formula calculation. We found that in small eyes (AL ≤ 22 mm), incorporating all available parameters was shown to be more significant to the IOL power selected. In contrast, the contribution of the optional parameters to the calculated IOL power is of little importance in eyes with AL more than 22 mm.

## Figures and Tables

**Figure 1 jcm-10-00542-f001:**
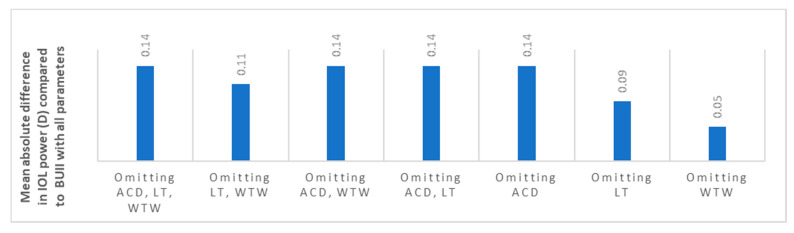
The mean absolute difference in IOL power calculation between partial biometry data and all Barrett Universal II (BUII) parameters in the whole cohort. ACD; anterior chamber depth, D; diopters, IOL; intraocular, LT; lens thickness, WTW; white-to-white.

**Figure 2 jcm-10-00542-f002:**
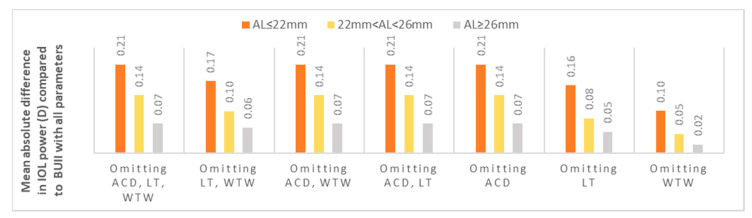
The mean absolute difference in IOL power calculation between partial biometry data and all BUII parameters stratified by axial length. ACD; anterior chamber depth, D; diopters, IOL; intraocular, LT; lens thickness, WTW; white-to-white.

**Figure 3 jcm-10-00542-f003:**
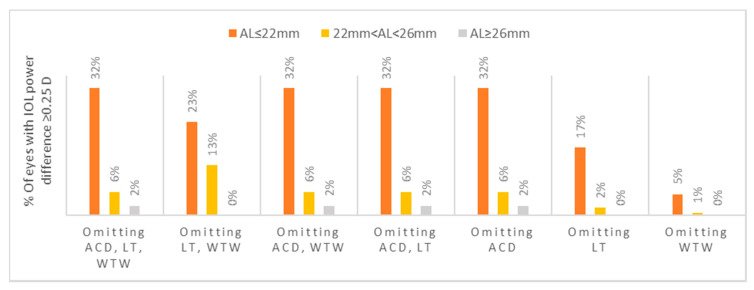
Percentage of eyes with a mean absolute difference of ≥0.25 D in the IOL power calculated between selected partial and all biometry data, stratified by axial length. ACD; anterior chamber depth, D; diopters, IOL; intraocular, LT; lens thickness, WTW; white-to-white.

**Figure 4 jcm-10-00542-f004:**
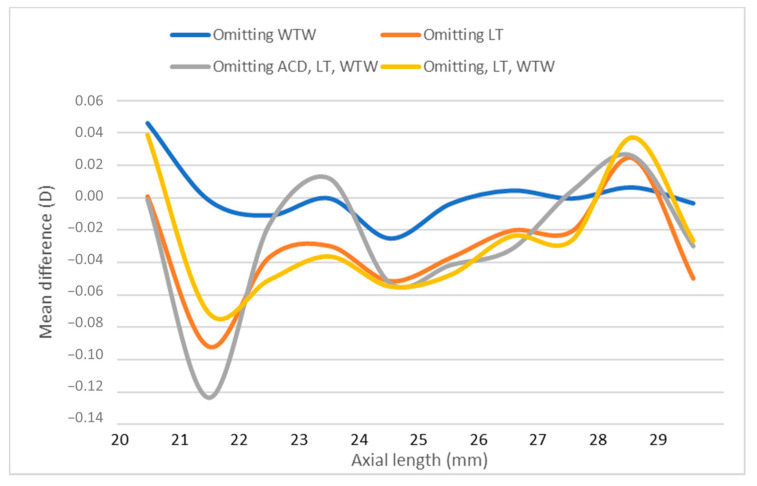
Smooth line graph of non-absolute difference in IOL power between selected partial and all data biometry in relation to axial length. ACD; anterior chamber depth, D; diopters, IOL; intraocular, LT; lens thickness, WTW; white-to-white.

**Table 1 jcm-10-00542-t001:** Clinical characteristics.

	Age (Years)	AL (mm)	K1 (D)	K2 (D)	ACD (mm)	LT (mm)	WTW (mm)
Mean	65.16	23.75	43.63	44.86	3.15	4.48	11.86
SD	17.38	1.88	2.03	2.07	0.48	0.49	0.50

ACD; anterior chamber depth, AL; axial length, D; diopter, K; keratometry, LT; lens thickness, WTW; white-to-white, SD; standard deviation.

**Table 2 jcm-10-00542-t002:** The mean predicted IOL power results of all various combinations.

Parameters Used	Mean Difference between Partial Biometry and All BUII Parameters (SD); CI; *p*-Value	Mean Absolute Difference between Partial Biometry and All BUII Parameters (SD); CI; *p*-Value
All BUII parameters	NA	NA
AL, K	0.03 (0.18); 0.01–0.05; <0.001	0.14 (0.12); 0.13–0.15; <0.001
AL, K, ACD	0.04 (0.13); 0.03–0.05; <0.001	0.11 (0.09); 0.10–0.11; <0.001
AL, K, LT	0.03 (0.18); 0.01–0.05; < 0.001	0.14 (0.12); 0.13–0.15; <0.001
AL, K, WTW	0.03 (0.18); 0.01–0.05; <0.001	0.14 (0.12); 0.13–0.15; <0.001
AL, K, LT, WTW	0.03 (0.18); 0.01–0.05; <0.001	0.14 (0.12); 0.13–0.15; <0.001
AL, K, ACD, WTW	0.04 (0.12); 0.03–0.05; <0.001	0.09 (0.08); 0.08–0.10; <0.001
AL, K, ACD, LT	0.00 (0.11); −0.01–0.01; 0.41	0.05 (0.09); 0.04–0.06; <0.001

ACD; anterior chamber depth, AL; axial length, BUII; Barrett Universal, CI; confidence interval, D; diopter, IOL; intraocular lens, K; keratometry, LT; lens thickness, SD; standard deviation, WTW; white-to-white.

## Data Availability

Data will be provided upon request.

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
