# Peer review of "High Agreement between Barrett Universal II Calculations with and without Utilization of Optional Biometry Parameters"

_jcm, 2021, doi:10.3390/jcm10030542_

Round 1

Reviewer 1 Report

Very interesting and innovative paper highlighting that optional variables, including a major one, ACD, have a small impact on IOL power calculation using the BUII formula, which would probably only be clinically significant in small eyes.

Nevertheless, in my opinion, one critical point should be recognized and addressed by the authors.

Line 145-150: since prediction errors were not calculated (no post-operative refractions were obtained) it is impossible from these results to infer "ACD, LT and WTW, optionally used by the Barrett calculator for enhancing accuracy, have more clinical significance in eyes shorter than 22 mm.".

While this is likely based on the author's beliefs (or mine or the formula's author), it is possible that the inclusion of these optional variables would worsen refractive results, particularly in this subgroup where IOL power was more likely to change. As the authors recognized in their literature review, we have no published evidence on this.

Stemming for this observation:

  • line 202 I suggest the authors acknowledge that since actual prediction errors were not calculated it is impossible to ascertain whether these variables would have a beneficial or detrimental impact
  • line 212 I suggest changing "desired" to a more neutral term
  • line 30-32 I suggest changing "optimal" to a more neutral term (or rewriting the conclusions of the abstract)

Author Response

Dear Professor Nelson Tansu,

Editor in Chief, Photonics

Thank you for your letter of January 14, 2021, encouraging us to revise our manuscript

 “High Agreement Between Barrett Universal II Calculations With and Without Utilization of Optional Biometry Parameters” (photonics-1074129)

We thank the Editorial Reviewers for their expertise and suggestions to improve our paper.  We have carefully read their comments and changed the manuscript accordingly.

We hope that this revised version is acceptable for publication in Photonics.

Our responses to the comments are enclosed. The manuscript has been changed accordingly.

Sincerely yours,

Uri Elbaz, MD,

Sackler School of Medicine,

University of Tel Aviv, Tel Aviv, Israel.

Department of Ophthalmology, Rabin Medical Center,

Schneider Children's Medical Center of Israel, Petah-Tikva, Israel

Very interesting and innovative paper highlighting that optional variables, including a major one, ACD, have a small impact on IOL power calculation using the BUII formula, which would probably only be clinically significant in small eyes.

Nevertheless, in my opinion, one critical point should be recognized and addressed by the authors.

Line 145-150: since prediction errors were not calculated (no post-operative refractions were obtained) it is impossible from these results to infer "ACD, LT and WTW, optionally used by the Barrett calculator for enhancing accuracy, have more clinical significance in eyes shorter than 22 mm.".

While this is likely based on the author's beliefs (or mine or the formula's author), it is possible that the inclusion of these optional variables would worsen refractive results, particularly in this subgroup where IOL power was more likely to change. As the authors recognized in their literature review, we have no published evidence on this.

Stemming for this observation:

  • line 202 I suggest the authors acknowledge that since actual prediction errors were not calculated it is impossible to ascertain whether these variables would have a beneficial or detrimental impact

We thank the Reviewer for this comment. We do acknowledge that the lack of reporting actual prediction errors is a limitation of our study. We therefore changed the Conclusions of the Abstract to focus on the IOL power selected prior to cataract surgery rather on the accuracy of the postoperative outcome and we added this as a limitation of our study.

The Conclusions of the Abstract were changed to the following (please refer to line 46 in our “tracked changes” manuscript):

“Using partial biometry data, the BUII formula in small eyes (AL≤22mm) resulted in a clinically significant difference in the calculated IOL power compared to the full biometry data. In contrast, the contribution of the optional parameters to the calculated IOL power is of little clinical importance in eyes with AL longer than 22 mm.”

The following was added to the limitations of the study at the end of the Discussion section (please refer to line 221 in our “tracked changes” manuscript):

“Furthermore, postoperative refractive data and prediction errors were not available and therefore we could not assess how the change in IOL power selected would influence the accuracy of the BUII. Of note, investigating the accuracy of the BUII was beyond the scope of our study and was shown previously in numerous studies. “

  • line 212 I suggest changing "desired" to a more neutral term

We removed the word “desired” and changed the sentence to the following (please refer to line 227 in our tracked changes manuscript):

“We found that in small eyes (AL≤22mm), incorporating all available parameters was shown to be more significant to the IOL power selected”.

  • line 30-32 I suggest changing "optimal" to a more neutral term (or rewriting the conclusions of the abstract)

The Conclusions of the Abstract were changed to the following:

“Using partial biometry data, the BUII formula in small eyes (AL≤22mm) resulted in a clinically significant difference in the calculated IOL power compared to the full biometry data. In contrast, the contribution of the optional parameters to the calculated IOL power is of little clinical importance in eyes with AL longer than 22 mm.”

Reviewer 2 Report

A short, simple, study well conducted with high level of potential interest to cataract surgeons

Author Response

A short, simple, study well conducted with high level of potential interest to cataract surgeons

We thank the Reviewer for this comment

Reviewer 3 Report

Excellent article, essentially flawless.

Author Response

Excellent article, essentially flawless.

We thank the reviewer for this comment.